# The Complexity of Bovine Leukemia Virus Oncogenesis

**DOI:** 10.3390/v17121609

**Published:** 2025-12-12

**Authors:** Florine Doucet, Alexis Fontaine, Malik Hamaidia, Jean-Rock Jacques, Thomas Jouant, Nour Mhaidly, Songkang Qin, Roxane Terres, Xavier Saintmard, Luc Willems, Manon Zwaenepoel

**Affiliations:** 1Molecular and Cellular Epigenetics (GIGA), University of Liège, B4000 Liège, Belgium; florine.doucet@uliege.be (F.D.); alexis.fontaine@uliege.be (A.F.); mhamaidia@uliege.be (M.H.); jean-rock.jacques@uliege.be (J.-R.J.); thomas.jouant@uliege.be (T.J.); nour.mhaidly@uliege.be (N.M.); songkang.qin@doct.uliege.be (S.Q.); roxane.terres@uliege.be (R.T.); xavier.saintmard@uliege.be (X.S.); manon.zwaenepoel@uliege.be (M.Z.); 2Molecular Biology (TERRA), University of Liège, B5030 Gembloux, Belgium

**Keywords:** enzootic bovine leukosis (EBL), bovine leukemia virus (BLV), human T-cell leukemia virus type 1 (HTLV-1)

## Abstract

Bovine leukemia virus (BLV) is a retrovirus infecting several bovid species, notably *Bos taurus*, where it fulfills Koch’s postulates for pathogenicity. The virus primarily targets B-lymphocytes, establishing lifelong infections that remain mostly asymptomatic but can progress to lymphocytosis or lymphoma. Transmission occurs through live infected cells via blood, milk, or transplacental routes. Despite a robust antiviral immunity, BLV replicates by producing virions (i.e., the infectious cycle) or inducing mitosis of infected cells (i.e., clonal expansion). The immune system effectively controls the infectious cycle but fails to impede clonal expansion, leading to chronic immune activation and immunosuppression. BLV modifies the transcriptome of the host cell by expressing oncogenic factors (Tax), viral microRNAs and antisense RNAs. Leukemogenesis arises from cumulative alterations of the virus (e.g., 5′-end deletions of the integrated provirus and histone modifications of the LTR promoter) and the host cell (e.g., genomic mutations and favorable chromatin integration). This model underscores a unique persistence strategy, linking chronic infection, immune evasion, and slow multistep oncogenesis in the bovine host.

## 1. The Spectrum of BLV-Infected Hosts Includes Different Bovid Species

Bovine leukemia virus (BLV) is a retrovirus that infects several species of bovids, including *Bos taurus* (cattle), *Bos indicus* (zebu), *Bos grunniens* (domestic yak), *Bos mutus* (wild yak), *Bubalus bubalis* (water buffalo), and *Bison bison* (bison) [1,2,3,4,5,6,7]. In bovines, BLV prevalence varies between 30% and 90% worldwide, with the notable exception of European Union countries, which are considered free of the disease, meaning that less than 0.2% of herds are infected. The nucleotide sequence of the different known BLV genotypes shows very little variation (<6%). Compared to HIV, the occurrence of quasispecies inside a host is limited, although mutations and deletions do occur. *Bos taurus* is the only species where the criteria of the Koch’s postulate associating a pathogen with his host are fulfilled: (i) BLV persists in the B cells of the diseased animals, (ii) the virus can be isolated from cultures or cloning, (iii) the provirus is pathogenic when introduced in a healthy host and (iv) the virus can be reisolated from the blood or tumors. Although epidemiological data indicates that other bovid species are also naturally infected by BLV, further evidence is needed to demonstrate the host’s spectrum. Besides *Bos taurus*, sheep have been considered to be strictly experimental models for BLV pathogenesis because the virus is not transmitted between hosts [8]. However, serological and molecular evidence suggests that BLV may also naturally persist in flocks of Iran and Colombia, although the mode of transmission is unknown [9,10,11].

In contrast to bovids and perhaps ovids, infection of humans by BLV and association with breast cancer is still disputed [12]. Indeed, different experimental approaches have led to divergent conclusions [13,14,15,16,17,18,19]. Parallel diagnostic analysis of samples in accredited laboratories is required to support the association of BLV and breast cancer. If this link is validated, further research is also needed to understand the modes of transmission and oncogenesis. In contrast to the related human T-cell leukemia virus (HTLV-1), the transmission and persistence of BLV in humans have indeed not been confirmed yet.

## 2. BLV Infects, Persists and Transforms B-Lymphocytes

The BLV viral particle consists of an icosahedral capsid surrounded by a matrix and a cellular membrane acquired during viral budding [2,5]. Viral envelope proteins embedded in this cellular membrane mediate interaction with the receptor and fusion with the target cell. The cellular receptor for BLV is the cationic amino acid transporter type 1 (CAT1/SLC7A1) [20,21]. Since this receptor is widely expressed, a broad variety of cell types can be infected with BLV in cultures. In vivo, however, the virus persists almost exclusively in B-lymphocytes and, to a lesser extent, in the monocyte/macrophage lineage [22,23,24]. BLV infects professional antigen-presenting cells (MHC-II+) that express the B cell receptor (BCR). The surface immunoglobulins (sIgM) of the BCR are associated with the negative regulator CD5 [25]. BLV-infected B lymphocytes circulate in the peripheral blood and migrate through the spleen and lymph nodes. Virus-infected cells can undergo affinity maturation and class switching (IgM → IgG) in germinal centers of secondary lymphoid organs [26]. The phenotype of the BLV target cell therefore corresponds to an immunocompetent (IgM+) or memory (IgG+) B-lymphocyte. The phenotype of the BLV-infected cell nevertheless remains imperfectly characterized, mainly due to limited availability of reagents and insufficient knowledge of bovine-specific B-cell markers [27].

In cattle, BLV persists lifelong in B-lymphocytes, leading to an asymptomatic and chronic infection in the large majority of cases. Approximately 30–50% of animals, depending on age and lifetime, will develop persistent lymphocytosis, characterized by a relatively stable increase in the B-lymphocyte counts. Between 5% and 10% of cases progress to leukemia or lymphoma within 3–7 years of infection [5,6,7,8,9,10,11,12,13,14,15,16,17,18,19,20,21,22,23,24,25,26,27,28]. Tumors most frequently develop inside lymph nodes (70%) and form lymphoma. Tumors may also affect the heart and abomasum and form extranodal lymphomas (also improperly called lymphosarcoma since a sarcoma is of mesenchymal origin). Tumors can lead to progressive wasting or sudden death, for example, due to splenic rupture and hemorrhage [29].

## 3. Viral Transmission Requires the Transfer of a Live Cell

Because the virion is highly unstable, BLV transmission requires transfer of a live cell infected with the virus (Figure 1). Perhaps the most frequent mode of transmission that accounts for the majority of the BLV-infected cases worldwide results from iatrogenic practices [30]. Despite veterinary best practices, the transfer of a few microliters of contaminated blood, predominantly by needle reuse, may be sufficient for BLV transmission. Additionally, the transplacental route from the cow to the fetus accounts for up to 15% of cases [31]. Later on, the virus can also be transferred from a BLV-infected cow to its calf through maternal milk [32,33]. While transmission from cow to calf is inhibited by passive immunity conferred by neutralizing antibodies present in the colostrum [34], calf infection by BLV becomes possible once passive immunity wanes, generally around 3–4 months of age. From this period onward, the virus transmitted from an infected host may replicate. Although only demonstrated experimentally, other modes of transmission may also include hematophagous insects (*Stomoxys*, *Culicoides*, tabanids) [35,36,37]. BLV can be isolated from blood-sucking insects, but effective transmission apparently requires interrupted feeding and immediate biting of a recipient host. This vector-driven mode has only been demonstrated experimentally when the insects were transferred by an experimenter. The use of fly nets has been associated with a decrease in herd prevalence, suggesting vector-driven transmission of BLV [38]. Since semen may contain viruses, BLV infection by mating is also possible but considered to be inefficient, in contrast to frequent transmission by sexual contact of HTLV-1 in humans [39].

Notwithstanding that all BLV-carriers are a potential source of contamination, the effectiveness of transmission primarily depends on the number of infected cells in the peripheral blood [40]. Selective removal of these virus shedders with high proviral loads can reduce prevalence in a herd [41,42,43,44]. Remarkably, this culling strategy based on abnormally high cell counts has been successful even before BLV was identified [45,46]. Besides the high viral loads, another parameter that correlates with transmission risk is the frequency of exposure.

## 4. BLV Undergoes Two Distinct Modes of Viral Replication

After transfer of the virus from an infected animal to a new host, BLV remains undetectable using serological (ELISA) and molecular (nested PCR) techniques. The virus is thus present in the animal but cannot be detected by currently available methods. The mechanisms of viral persistence during this early period and the phenotype of the infected cell are unknown. It is conceivable, however, that BLV-infected maternal cells cross the placenta and persist in the fetus at least temporarily by microchimerism. Alternatively, it is likely that phagocytes from the recipient host (e.g., resident macrophages surrounding the needle prick) engulf virus-infected cells before initiating the host’s immune response. A delay in this process, which can last from 1 week to several months [47], may explain the emergence of seronegative animals even when they have been isolated from BLV-infected herds.

After this initial phase, the BLV genomic RNA is converted into DNA by the virally encoded reverse transcriptase [48]. The newly formed provirus then stably integrates into the genome from the recipient host. After insertion into the B-cell chromosome, the provirus may remain transcriptionally silent or produce new viral particles. At these early steps, provirus integration occurs randomly in accessible regions of the chromatin [49]. This period is followed by very active viral replication and the development of a humoral and cytotoxic immune response (Figure 1). Therefore, the vast majority of infected cells will be destroyed by the host’s immunity. The infected cells that survive this massive depletion will then undergo mitotic division and expand to form clones (i.e., clonal expansion) [50,51]. The surviving clones are located in transcriptionally active sites of the chromatin, suggesting the involvement of viral factors in driving cell mitosis (e.g., gene activation of cellular genes *in trans* by the Tax oncogene or *in cis* by proximal integration of the provirus). Despite the development of a strong immune response, the host is nevertheless unable to clear the virus. Once infected, cattle indeed remain carriers of the virus for life [5]. The infection is chronic but not latent because the immune response does control viral replication and prevents infection of new cells. It is remarkable that cells carrying an integrated provirus can persist throughout the animal’s life. Indeed, cell clones present in tumors were shown to have emerged during early stages of infection that correspond to the few weeks that follow the transmission of the virus to the host [52].

BLV can thus replicate either by producing viral particles or by inducing mitosis of its host cell despite a highly effective immune response. The mechanisms driving clonal expansion of infected cells are still incompletely understood but involve viral oncogenic proteins and noncoding RNAs [53,54,55].

## 5. The Host’s Immune Response Controls Viral Replication

There is an apparent paradox between the nearly undetectable expression of the virus and the maintenance of a strong antiviral immune response in infected animals. High levels of antiviral antibodies and the presence of active cytotoxic T-cells nevertheless imply that the virus is expressed. However, in situ RNA hybridization shows that almost all BLV-infected cells are transcriptionally silent [56]. It is possible that plus-strand viral RNAs are transcribed at levels below the limit of detection or, alternatively, that transient bursts of expression are quickly turned off, as demonstrated for HTLV-1 [57]. Isolating BLV-infected cells by venipuncture followed by a brief cell culture allows abundant expression of viral proteins [58]. This process can be interpreted as re-activation of viral expression in the absence of a putative plasma inhibitory factor [59]. However, BLV expression is induced in blood even without the removal of plasma [60]. Therefore, a more likely hypothesis is that the virus continuously attempts to express viral proteins and that the host immune response very effectively destroys BLV-infected cells [61].

The role of the immune response in continuously controlling the virus is supported by numerous studies [8]. Experimental evidence shows that:-Particular alleles of the β chain of the class II major histocompatibility complex (BoLA DRB3) are associated with low viral loads [41,42,62,63,64,65]. Thus, allelic variations in MHC-II can influence viral replication and disease progression.-Lymphoid organs such as the spleen regulate the onset of leukemia [66]. Splenectomized sheep develop disease faster.-Cells expressing the virus have a shorter lifespan [67]. Transient stimulation of viral expression ex vivo reduces persistence of infected cells after reinfusion in vivo.-Improving promoter strength of the LTR paradoxically reduces BLV replication [68].-Activation of viral expression with a histone deacetylase inhibitor reduces proviral loads and has therapeutic potential [69,70].-It is very difficult, if not impossible, to inoculate one BLV strain into an animal already infected with another genotype. This concept has led to the development of a live-attenuated vaccine capable of effectively protecting animals in herds with high prevalence [71,72].

Together, these findings reveal a robust immune response capable of controlling viral replication but unable to eradicate the virus. Due to the permanent stimulation of antiviral immunity, the expression of immune checkpoints, in particular PD-1 and TIM-3, are increased on T-lymphocytes during tumorigenesis [73]. This T-cell exhaustion leads to impaired immunity of infected animals [74,75,76,77,78]. Thus, BLV infection is associated with immunosuppression, favoring opportunistic infections such as mastitis [79,80,81,82], poor response to treatments against bacteria, parasites and chronic diseases (e.g., pneumonia, gastroenteritis) [83] and reduced vaccine efficacy against other pathogens (such as BHV-1 or rotaviruses) [84,85,86].

## 6. Despite Host’s Immunity, BLV Persists Lifelong Leading to Leukemogenesis

Considering the number of BLV-infected cells and their turnover, the onset of leukemia/lymphoma is relatively infrequent in the bovine species (between 5% and 10% of cases). Similarly, the long latency time preceding leukemogenesis (3–7 years) indicates that the virus is not very oncogenic. As a *deltaretrovirus*, BLV lacks sequences homologous to the bovine genome but is characterized by a region X encoding regulatory genes [87,88,89,90,91]. Among these, the Tax protein, a transcriptional regulator of viral and cellular expression, displays oncogenic potential in standard transformation assays [53]. The X region further encodes viral microRNAs that promote persistence and replication by targeting cellular genes such as c-fos, granzyme B, PPT1, and HMCN1 [54,55,92,93]. The antisense strand can also be transcribed from a promoter located in the 3′LTR [94,95], similarly to the HBZ gene of HTLV-1 [96]. The BLV antisense RNAs are consistently expressed during infection, modifying host signaling pathways and epigenetics [97,98]. Antisense RNAs may even extend into the host genome and alter the expression of cellular genes at a distance [99].

Due to the chronicity of the infection, however, it is clear that these noncoding RNAs and oncogenic factors (Tax) are insufficient to induce leukemia/lymphoma (Figure 2).

Therefore, the leukemogenesis process involves multiple non-exclusive mechanisms that include:-(i) Genetic mutations inside the integrated provirus. Base substitutions and duplications appear in the provirus during leukemogenesis [100,101]. Limited genetic variability is concentrated in envelope and LTR promoter regions (10% at the nucleotide level). Large deletions may occur, generating noninfectious proviruses. The process of homologous recombination generates type 1 deletion mutants lacking the 5′ region of the viral genome. Although this process is frequent at late stages of the pathogenesis, a complete and infectious proviral copy remains intact in the infected animal [102].-(ii) Epigenetic modifications. Histone deacetylation of LTR chromatin locks down viral transcription [95]. Cytosine methylation of the LTR promoter reduces and shuts off viral gene expression. These mechanisms allow the virus to remain silent and undetectable by the host immunity.-(iii) Integration into favorable chromatin sites. Although integration is random, only proviruses in favorable chromatin zones persist [49,103]. Integration in these sites initiates transcription of flanking cellular genes [99].-(iv) Mutations of the host cell genome. Uncontrolled expansion of infected cells leads to mutations in the cellular genome, which may contribute to leukemogenesis [104,105,106]. Enforced clonal expansion of infected cells is associated with the onset of mutations, such as those affecting p53, promoting tumor development.

## 7. Conclusions

The model of viral persistence shows that BLV remains active throughout the host’s life by promoting division of infected cells. Provirus-carrying B lymphocytes renew at a faster rate than healthy cells, contributing to genomic mutations and disease progression. Although the immune response is vigorous, it cannot eliminate the virus, which relies on a series of sophisticated strategies, notably based on noncoding RNAs.

## Figures and Tables

**Figure 1 viruses-17-01609-f001:**
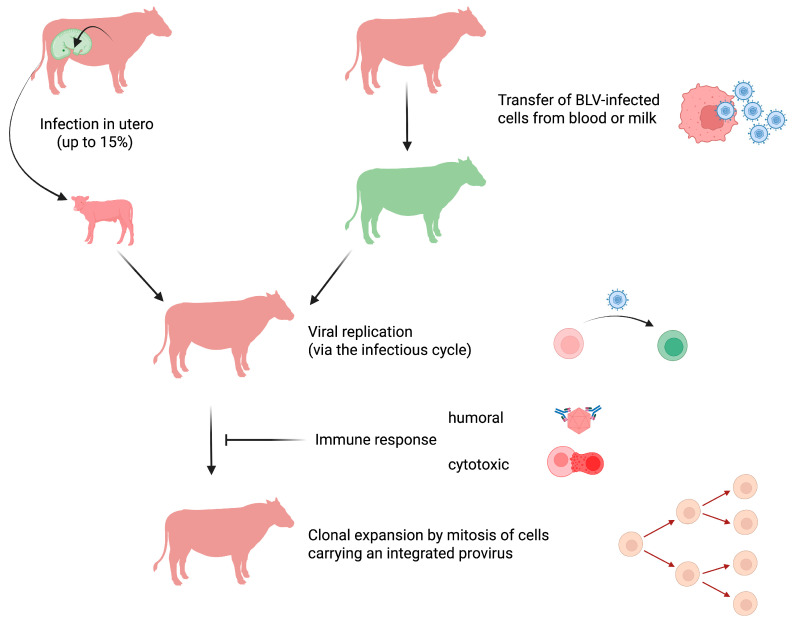
**BLV transmission and modes of replication.** In utero BLV transmission from cow to calf occurs in up to 15% of cases (non-infected with BLV in green; infected with BLV in red). Transfer of live cells containing an integrated BLV provirus from an infected to a healthy animal accounts for most viral transmissions. BLV-infected cells present in the blood and milk are transmitted perinatally, through iatrogenic routes (e.g., reuse of contaminated needles), and possibly vectors (e.g., insects). Shortly after the transfer of a BLV-carrying cell, the virus actively replicates through the infectious cycle. Viral expression triggers a strong humoral and cytotoxic immune response that destroys the majority of BLV-infected cells. Only a small proportion of BLV-infected cells survive and replicate mainly through mitotic division. This clonal expansion allows for virus persistence and ultimately leads to persistent lymphocytosis and EBL. Image created by Biorender.com.

**Figure 2 viruses-17-01609-f002:**
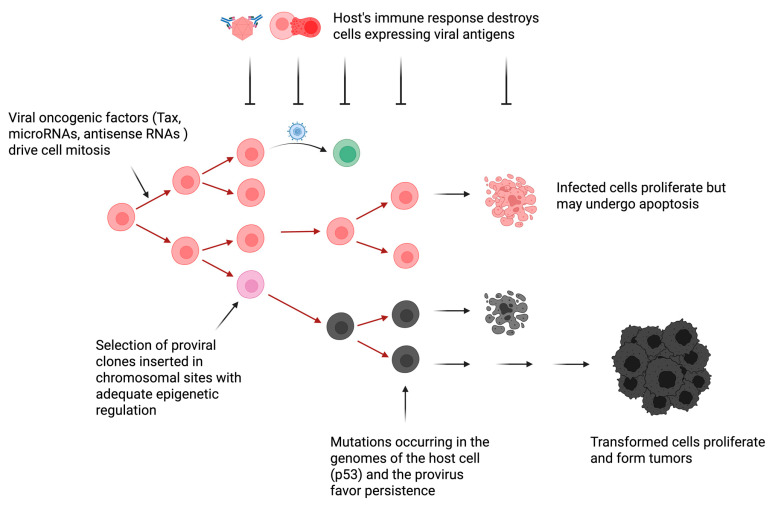
**Hypothetical model of BLV-induced oncogenesis.** The mechanisms driving proliferation of a BLV-infected cell are still imperfectly known but include the expression of viral oncogenes (e.g., Tax) and RNAs (e.g., BLV-encoded microRNAs and antisense RNAs AS1 and AS2). Initial provirus integration is believed to occur randomly into exposed sites of the chromatin. Most of these clones are destroyed upon the onset of the host’s immune response. Only a limited number of clones inserted in chromosomal sites characterized by an adequate epigenetic regulation survival. These chromosomal sites allow silencing of sense transcripts encoding viral proteins (e.g., capsid, matrix and envelope), a transient burst of expression of oncogenic factors (e.g., Tax) and transcription of BLV micro/antisense RNAs able to modify the cell transcriptome. Compared to uninfected cells, B lymphocytes carrying a BLV provirus undergo a faster turnover, providing that proliferation rates exceed apoptosis. If not detrimental, mutations in the host genome (e.g., p53) or in the provirus (e.g., deletions of the 5′LTR) may further promote persistence and proliferation of the infected cell. This complex and dynamic process occurs under the permanent control of the host’s immunity. After a long latency period, leukemia/lymphoma appears when this equilibrium is disrupted. Image created by Biorender.

## Data Availability

No new data were created or analyzed in this study. Data sharing is not applicable to this article.

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
