# Peer review of "The Complexity of Bovine Leukemia Virus Oncogenesis"

_viruses, 2025, doi:10.3390/v17121609_

Round 1
Reviewer 1 Report
Comments and Suggestions for Authors
This is a well written review that provides an interesting compilation of knowledge, competently selected from publications, to present a multilevel mechanism of Bovine Leukemia Virus induced oncogenesis.
I have some minor comments.
Line 92 – why Authors did not mentioned Culicoides?
Line 94 – how do Authors understand sentence: this vector-driven mode has only been demonstrated experimentally?
Any comments for the papers: M. Ooshiro et al. M. https://doi.org/10.1136/vr.101833 or Junko Kohara et al. https://doi.org/10.1292/jvms.18-0199
Line 137 – Please expand and clarify sentence : The surviving clones are located in transcriptionally active sites suggesting the involvement of viral factors in driving cell mitosis.
Is this related to the site of virus integration?
Line 140 -143: The infection is chronic but not latent because the immune response does control viral replication and prevents infection of new cells. It is remarkable that cells carrying an integrated provirus can persist throughout the animal’s life. Indeed, cell clones present in tumors were shown having emerged during early stages of infection .
Please clarify : So, is that how it is in the first phase?
Authors’ Contributions are missing….
Author Response
This is a well written review that provides an interesting compilation of knowledge, competently selected from publications, to present a multilevel mechanism of Bovine Leukemia Virus induced oncogenesis.
Thank you.
I have some minor comments. Line 92 – why Authors did not mentioned Culicoides?
"Culicoides" is now mentioned.
Line 94 – how do Authors understand sentence: this vector-driven mode has only been demonstrated experimentally?
There is indeed no direct evidence demonstrating that insects transmit BLV in natural conditions (i.e., without interruption of the feeding and transfer to another animal by an experimenter).
Any comments for the papers: M. Ooshiro et al. M. https://doi.org/10.1136/vr.101833 or Junko Kohara et al. https://doi.org/10.1292/jvms.18-0199
We added the information of these papers showing that the use of fly nets correlated with a decrease of BLV prevalence.
Line 137 – Please expand and clarify sentence : The surviving clones are located in transcriptionally active sites suggesting the involvement of viral factors in driving cell mitosis. Is this related to the site of virus integration?
We clarified by adding on line 139 : "... The surviving clones are located in transcriptionally active sites of the chromatin suggesting the involvement of viral factors in driving cell mitosis (e.g., gene activation of cellular genes in trans by the Tax oncogene or in cis by proximal integration of the provirus)..."
Line 140 -143: The infection is chronic but not latent because the immune response does control viral replication and prevents infection of new cells. It is remarkable that cells carrying an integrated provirus can persist throughout the animal’s life. Indeed, cell clones present in tumors were shown having emerged during early stages of infection. Please clarify : So, is that how it is in the first phase?
We specified that the early stage of infection corresponds to the few weeks that follow the transmission of the virus into the host.
Authors’ Contributions are missing….
The authors contributions are now included.
Thank you for your helpful comments
Reviewer 2 Report
Comments and Suggestions for Authors
This is a well written review. Here are a few minor comments.
- The three acronyms as Key words should be spelled out, particularly EBL, which is not defined in the Abstract.
- Page 2. Lines 72/73. Ref. 29 is focused on the impact of bovine leukemia virus infection on beef cow longevity, not showing the data of 5-10% infected cases progress to leukemia or lymphoma in 3-7 years as the authors cited.
- Any information of the lymphoma types associated with BLV? High or low grade? And the types of leukemia?
- Are there any specific reasons or pathways to explain why BLV persists in B lymphocytes, but not T lymphocytes like HTLV-1?
Author Response
This is a well written review. Here are a few minor comments.
Thank you.
- The three acronyms as Key words should be spelled out, particularly EBL, which is not defined in the Abstract.
Acronyms have been added.
- Page 2. Lines 72/73. Ref. 29 is focused on the impact of bovine leukemia virus infection on beef cow longevity, not showing the data of 5-10% infected cases progress to leukemia or lymphoma in 3-7 years as the authors cited.
Reference #29 has been deleted and replaced by #5
- Any information of the lymphoma types associated with BLV? High or low grade? And the types of leukemia?
To our knowledge, this information is unfortunately not available.
- Are there any specific reasons or pathways to explain why BLV persists in B lymphocytes, but not T lymphocytes like HTLV-1?
The answer to this interesting question is unknown.